# The Environmental Impact and Formation of Meals from the Pilot Year of a Las Vegas Convention Food Rescue Program

**DOI:** 10.3390/ijerph16101718

**Published:** 2019-05-16

**Authors:** Samantha To, Courtney Coughenour, Jennifer Pharr

**Affiliations:** 1Coconino County Public Health Services District, Flagstaff, AZ 86001, USA; 2School of Public Health, University of Nevada Las Vegas School of Public Health, Las Vegas, NV 89154, USA; Courtney.Coughenour@unlv.edu (C.C.); Jennifer.Pharr@unlv.edu (J.P.)

**Keywords:** food rescue, convention center, greenhouse gas emissions, food security, landfill diversion

## Abstract

Annually, millions of tonnes of leftover edible foods are sent to landfill. Not only does this harm the environment by increasing the release of greenhouse gases which contribute to climate change, but it poses a question of ethics given that nearly 16 million households are food insecure in the US, and hundreds of millions of people around the globe. The purpose of this study was to document the amount of food diverted from landfill in the pilot year of a convention food rescue program and to determine the amount of greenhouse gas (GHG) emissions avoided by the diversion of such food. In the pilot year of the convention food rescue program 24,703 kg of food were diverted. It is estimated that 108 metric tonnes of GHG emmisions were avoided as a result, while 45,383 meals for food insecure individuals were produced. These findings have significant implications for public and environmental health, as GHG emissions have a destructive effect on the earth’s atmosphere and rescued food can be redistributed to food insecure individuals.

## 1. Introduction

‘Food waste’ is defined as any edible item that goes uneaten at the retail or consumer level [1]. An estimated 31% of all food sold and prepared at these levels is ultimately wasted [1]. Reasons for this waste include over-purchasing by both retailers and consumers, and over-sized portions [2]. This often results in the over preparation of food for meals, which results in food waste [3,4]. 

Wasted food is one of the largest contributors to landfill in United States (US) and has major environmental and economic impacts. A recent report by the US Environmental Protection Agency (EPA) stated that of the municipal solid waste sent to landfill in 2014, 29 million tonnes, or 21.6% of all waste, was discarded food. This is a concern, as by-products have the potential to contaminate land, water, and air [5]. Greenhouse gases (GHG) released by food decay contribute to climate change, increasing global surface temperatures, and have the potential to effect health and wellbeing around the world [6]. GHG include carbon dioxide (CO_2_), methane (CH_4_), and nitrous oxide (N_2_O), which combined are calculated as CO_2_ equivalents (CO_2_E) [6]. The continual release of GHG into the atmosphere increases solar energy absorption, and the global temperature cyclically climbs [7], effecting animal and plant biodiversity, reducing freshwater resources, worsening acidification of oceans, and depleting the ozone layer [8]. Impacts on human health include more natural disasters, famine-induced malnutrition, and severe mortality rates during extreme weather events [8]. Positive associations have been reported between climate change and a rising risk of infectious diseases [8], especially vector-borne and water-borne illnesses [9]. Mosquito-borne diseases, cholera, and salmonellosis outbreaks are expected to amplify with increased atmospheric temperatures; even seasonal allergies will surge with extended pollen seasons [9]. With such severe implications to environmental and human health, approaches to prevent GHG emissions and the resultant damage to the atmosphere should be paramount and incorporated into all industries. 

The EPA reports that in the US, the decaying of solid waste in landfill accounts for 34% of human-related methane production [10], 16% of which is from uneaten food alone [7]. Tonini and colleagues estimate the global warming impact of avoidable food waste in the United Kingdom (UK) to be between 2000 and 3600 kilograms (kg) CO_2_E per tonne of food waste [11], while Salemdeeb estimated that preventing avoidable food waste in the UK could lead to a reduction in GHG by a magnitude of 706 to 896 kg CO_2_E per tonne of food waste [12]. It is also estimated that, in the US, 1.4 kg of CO_2_E per person per day are produced by food wasted at the consumer and retail levels, excluding food that is nonedible. This amount of CO_2_E produced from food waste is equal to the annual emissions of 33 million average passenger vehicles [13]. 

Additionally, it is estimated that the economic impact of food waste globally is 750 billion (US) dollars per year [14,15], while in the US it is nearly 200 billion dollars [10,13]. The economic cost of food waste varies based on the size and state of development of the country, but some of the estimates are 7.7 billion (US) dollars per year in South Africa [16], 18.3 billion in the UK [17], 21.1 billion in Canada [18], and 5.8 billion in Australia [19]. Preventing the production of CO_2_E and the wasting of economic resources by diverting food waste is a controllable way to reduce the impact of global warming and food insecurity [7,20]. 

Food insecurity is defined as the “limited or uncertain availability of nutritionally adequate and safe foods, or [the] limited or uncertain ability to acquire acceptable foods in socially acceptable ways” [21]. In 2015, it was recorded that 15.8 million households (12.7%) in the US were food insecure [22], a portion of the over 700 million undernourished people globally [15]. At any point of a given year, about one in ten Americans will experience food insecurity, while approximately half of all children will, at some point during their childhood, live in a household that requires food assistance [21,23]. 

Being food insecure is not always consistent. In some households, the food intake and eating patterns of one or more members is reduced or disrupted at various times of the year due to a lack of money or resources [10]. An investigation into this subject reveals that households are not lacking food on all days of the year, but experience insecurity for around seven months at a time [24]. 

Annual income and lack of financial management skills can play an important role in the level of food insecurity a household experiences, but other risk factors that have been associated include being at risk of homelessness, a household head who is American Indian, not receiving child support, a noncustodial father that does not regularly visit, seasonality, residing in a state with unemployment rates higher than average, and having at least one cigarette smoker in the home [25]. A report by Feeding America noted that food-insecure families reported altering eating habits in order to afford non-food items, raising concerns about the potential health risks to families with children [26]. Surveys revealed that families placed less importance on paying for food and medication and prioritized paying for rent, water and utility bills, and transportation costs [26]. In one US study of the households surveyed who had cut back on food, 24% did so to afford basic household goods, such as toothpaste, laundry detergent, diapers, or shampoo [26].

A lack of adequate food and poor nutrition have several negative impacts on the physical and mental health of children, adults, and seniors. Food insecure mothers-to-be are more likely to suffer from oral health and mental health problems, as well as increased risk of birth defects. Negative physical health outcomes including exacerbated chronic illnesses, oral health problems, increased risk of asthma, frequent stomachaches, headaches, and iron deficiencies, as well as adverse mental health impacts including behavioral troubles, more visits to a psychologist, and impaired intellectual proficiencies occur in children who experience food insecurity [27]. Adults in food-insecure households are more likely than their food-secure counterparts to experience long-term physical health problems, higher levels of depression, and to have more advanced levels of chronic disease [28]. In senior adults, food-insecurity is associated with higher rates of diabetes, hypertension, heart disease, asthma, depression, functional impairment, limitations to activities of daily living, and a lack of social support [24,29]. These negative health outcomes influence health care costs and hospitalizations, leading to job instability, and a lower quality of life [30,31]. Increasing access to wholesome, nutritious foods using foods that would have been discarded could be an opportunity to positively affect the health and well-being of food-insecure individuals.

Allowing edible food to decompose in landfills should be reconsidered when so many people suffer from hunger and food insecurity every year. An estimated 31% of all the food sold and prepared at retail and consumer levels is wasted annually in the US [1]. Food surplus is routinely generated by food retailers due to the seasonality, variability, and unpredictability of customer demand, however a notable reason for such high rates of restaurant waste is due to large portion sizes. Restaurants entice customers with larger portion sizes to convey that the payer is receiving a bargain; however, this trend creates more waste when the customer cannot finish the meal [3]. Trends of large meals encourage a culture of overconsumption and waste [3]. The EPA food recovery hierarchy prioritizes collecting unspoiled, healthy food to redistribute to hungry people, second only to source reduction [2]. While strides have been made in the US to reduce food waste, it is improbable that this will prevent it entirely [10]. The most frequently promoted solution to such waste focuses on the management of the existing surplus through efficient recovery [32], creating an appropriate setting to incorporate food recovery practices. 

Las Vegas, Nevada is a hub for convention gatherings, accommodating over six million visitors per year [33]. Aria, a major resort and convention center owned by MGM Resorts International, realized the potential of its untouched convention food to improve both community wellbeing and environmental health. Hotels and convention centers such as this one take pride in and attract endless clients by providing lavish banquets resources, including quality food. As a result, meals prepared for convention attendees are overestimated to ensure that sufficient portions can be offered to each guest. This overabundance of high-quality food inspired a pilot food rescue program between Aria and a local food bank. 

Aria’s convention center offers corporations a selection of event spaces that can accommodate over five thousand attendees at a time [34]. To ensure the efficacy and safety of the program, a partnership between the MGM Resorts International, the local health district, and Three Square Food Bank (a member of Feeding America) developed policies and protocols to ensure that when an abundance of food had been prepared for conventioneers, it was suitable for donation. Events that have more than 500 attendees are expected to generate enough surplus meals to initiate the food rescue program process [35]. The surplus food that has not left the kitchen and has maintained a safe temperature in the hot box, in its original tray, never having been served to a guest gets donated [36]. A team from the Three Square Food Bank arrives to the convention center with a refrigerated truck, tests the contents to confirm that the temperature and quality of ingredients are safe, and the leftover foods are then transferred to trays provided by the food bank. The quantity is catalogued, and then transported to a Three Square Food Bank warehouse in hotboxes to maintain appropriate temperature [35]. At the warehouse, the trays are labeled and placed into a blast chiller and stored in freezers [35]. Details of the recovered products (what types of food, how many palettes, date available) are entered into an inventory system, from which partners of the food bank can browse, select, and order food items to be used at their sites in congregate meals. Meals are distributed to clients by a network of charitable, nonprofit organizations [37]. 

This program is unique because, to the best of our knowledge, food rescue efforts of edible, prepared convention food has not been executed on such a large scale before. The purpose of this study was to document the amount of food in pounds diverted from landfill in the pilot year of the convention food rescue program, estimate the number of meals created from the diverted food, and determine the amount of GHG emissions avoided by the diversion of such food.

## 2. Materials and Methods 

### 2.1. Rescued Food

This study quantified the amount of food recovered in pounds from Aria’s convention center during the pilot year (August 2016 to July 2017). The amount was determined by weighing each tray of food prior to being transported from the Aria and chilled at Three Square Food Bank’s warehouse, and then donated to food pantries. Using the United States Department of Agriculture’s (USDA) estimate that each meal consists of about one half (0.544) kg of food, the number of meals provided to food-insecure individuals in Las Vegas, NV by agency partners from this rescued food was estimated [38]. 

### 2.2. GHG Emissions Avoided

The estimated amount of GHG emissions avoided due to the redirection of food was determined using the EPA’s Waste Reduction Model (WARM) [39], a tool provided by the EPA to calculate the change in amount of GHG saved by utilizing alternate scenarios. Prevented GHG emissions are calculated by comparing the emissions associated with the baseline material management (i.e., current practices) with an alternative scenario, as opposed to simply multiplying the quantity of materials managed by an emission factor [39]. The EPA has conducted extensive research to determine “the life-cycle GHS and energy factors for materials across several categories (e.g., plastics, metals, woods [… food waste])”. This life cycle varies based on many factors related to the local landfill operations. The WARM program incorporates the conditions under which the specific landfill operates, which are outlined below. 

The total weight of rescued food was used as the baseline amount of waste generated, and entered as food waste, tonnes landfilled in step 1 of the WARM program. The alternative management scenario for the current study was source reduction, as the waste would have been diverted from landfill; the total weight of rescued food was again entered as food waste, tonnes source reduced in step 2. Parts 3 through 9b are related to the conditions under which the specific landfill operates. The landfill that the food would have been transported to is Apex Landfill. This landfill is managed by Republic Services, occupies over 2200 acres and is located in Lincoln County, NV [40]. The specific conditions under which this landfill operates are as follows: this region is considered a mountainous region in the WARM program. The food waste is not a product of recycling and is therefore considered virgin. Landfill gas is recovered at this site with typical efficiency and is used for energy [41]. The decay rate of this landfill is most accurately described as moderate and dry [41]. Food waste digestion is not done separately from the landfill cell and therefore is not cured. Lastly, WARM incorporates emissions released during transportation to the waste management facility in an attempt to offset the CO_2_E created by the garbage haul process; it is 47.3 km between the Aria convention center and the local landfill. From the calculated CO_2_E, proportions of CO_2_, CH_4_, and N_2_O were determined based on percentages published in previous research [6].

## 3. Results

The amount of convention food rescued was collected by Three Square Food Bank by weighing the amount of food brought to their warehouse to be chilled. From August 2016 through July 2017 the program had yielded 24,703 kg of donated food [38]. Obtaining the exact number of meals created from the rescued food was not possible since agency partners are not required to weigh the distributed portions served at congregate meals. However, using the USDA’s estimate that each meal consists of about 0.544 kg of food [42], it is estimated that 45,383 meals were produced from donated convention food in the 12-month time frame.

The pilot year of the convention food rescue program yielded 24,703 kg, or 24.7 metric tonnes, of donated food. This was used as the baseline amount of waste generated and source reduction was entered as the alternative management scenario, as the waste that was diverted from landfill. The WARM program estimated that by diverting 24,703 kg of food waste from the landfill, the resulting change in GHG emissions were 108 metric tonnes (MT) of CO_2_E avoided annually (see Table 1). A GHG equivalencies calculator determined that 108 MT of CO_2_E is equivalent to GHG emissions from 22.9 passenger vehicles driven for one year, CO_2_ emissions from 46,004 liters gasoline consumed, 16.2 homes’ annual electricity use, 250 barrels of oil consumed, or 53,597 kg of coal burned [43] (See Figure 1).

The proportions of CO_2_, CH_4_, and N_2_O are reported to be 76.7%, 14.4%, and 7.9% of CO_2_E, respectively [6]. The ratios of carbon dioxide, methane, and nitrous oxide prevented by diverting convention food rescue program food waste from Apex Landfill can be viewed in Table 1.

## 4. Discussion

Addressing food waste by incorporating a rescue program into a food supply chain is a responsible method of promoting a sustainable food system. Building collaborations between organizations that reduce food waste while increasing access to adequate meals is an important initiative [44]. Our study shows that a convention center food rescue program’s pilot year resulted in avoidance of GHG emissions and provided many meals for food insecure individuals.

Food waste is a public health concern due to environmental implications, and various methods to offset waste are considered and executed. Apex Landfill has made state-of-the-art strides towards changing the environmental impact it has on the community. The institution presently collects methane from the landfill and sends the gas to a power plant where it is used to produce enough electricity to power over 9000 homes in the Las Vegas Valley [41]. Composting is not a current option offered by the landfill: however, some food waste that is transported from the Las Vegas Strip is consolidated separately at the landfill and is used to supply feed for hogs at a neighboring pig farm. Matured pigs are available for sale to neighboring cities, and future intentions at the site include the incorporation of hog bedding into composting mounds, from which nutrient-rich soil will feed on-site plant nurseries where fruit and vegetables will be grown and offered to casino restaurants [41]. While the intentions are reasonable, livestock manure is the fourth largest source of methane production, just behind landfills, with an increasing trend of emissions from swine and dairy cow [45]. These efforts by Apex and other regional landfills to reduce the environmental impact are genuine, methane will continue to be produced as a byproduct from the animals. Therefore, while diverting food scraps to animal farms is an option in the EPA’s food recovery hierarchy more superior to landfilling and composting, source recovery and feeding hungry people is still most preferred, especially with surplus edible food products. Researchers have found that food rescue was “more economically costly than landfill or composting, [though it] is a lower cost method of obtaining food for the food insecure than direct purchasing” [46]. Additionally, Phillips et al. suggest that the “costs [of food rescue] can be reduced (and supply increased) simply by recruiting additional donors to participate [47].” This is noteworthy for the current study, as many additional convention centers are interested in becoming involved in food rescue efforts.

Redirecting nutritious foods to populations in need may result in positive changes in their health and well-being. Because food costs influence everyday spending, a program that has the potential to impact a household’s food security status, such as congregate meal sites, has the potential to impact health and wellbeing [27]. Participating in food assistance programs has been shown to liberate funds that might have been spent on food, reallocating resources to be spent on other costs such as health care, strengthening a positive impact on inhabitants’ health, and increasing independence among struggling families [27]. Similarly, the stress caused by the lack of monetary ability to provide food to their family increases maternal stress and may impact mental health status. Researchers explain that causality of food insecurity and poor health conditions may go in both directions; for example, limited nutrition intake in the food-insecure could lead to exacerbation of chronic conditions such as diabetes or HIV, while having diabetes or HIV and the resulting medical costs may trigger someone to be food insecure [48]. Access to no-cost, nutritious food through food assistance programs can relieve some of these stressors. Additionally, improving nutritional intake could break the cycle, eliminating or minimizing the negative health outcomes and alleviating costs of long-term medical care.

Increasing access to nutritious meals have beneficial effects on the health and well-being of individuals of all ages. Living in a food secure household can positively impact on a child’s health [27], as well as make a difference in the child’s academic performance and social skills [49]. Also, if single mothers are more food secure, they can be more financially secure and independent from damaging, non-custodial partners. The liberation from a negative influence removes numerous other factors that are detrimental to the growth of their children [50].

The majority of food waste in high income countries occurs at the consumption stage, therefore the implementation and expansion of programs similar to Aria’s convention food rescue program are necessary and appropriate [51]. Research, such as that done by Raji et al. has shown that limited freeze-thaw cycles have limited nutritional loss [52]. Conversations with convention and food bank stakeholders and compliance to best practices ensure the safety and nutritional quality of the meals that have been chilled, frozen, and reheated once, and should be encouraged elsewhere. 

The convention food rescue program is just beginning and not yet performing at full capacity. Current efforts are underway to expand the convention food rescue program and landfill diversion practices to other convention centers at various casino properties. Considering that Las Vegas is one of the leading convention destinations in the US, expansion of convention food rescue in Las Vegas, alone, has the ability to contribute significantly to reductions in GHG. It is predicted that the convention food rescue program will set precedence for other large-scale food establishments and convention facilities, both within and outside of the Las Vegas Valley. Increasing the amount of recovered food would prevent future food waste degradation from releasing tonnes of harmful GHG into the atmosphere. Climate change is a critical, all-encompassing public health issue and approaches to reduce the emissions that harm the atmosphere should be explored.

Convention food rescue programs can be used in conjunction with other food waste prevention strategies and policies to reduce food waste’s contribution to GHG emissions. These strategies include (1) food redistribution policies to reattribute edible retail and commercial food to food banks, (2) education of the public and commercial/retail food employees about food waste, portion size, food purchasing and planning, and food preparation, (3) logistical improvements, such as serving plated meals rather than buffet-style [14]. Potential food waste prevention policies could be tax incentives for food donation, limited liability regulations for donors, logistical support for the collection and transportation of edible food, tax incentives for the prevention of food waste, and support for research and development of new innovative initiatives to reduce food waste [14]. 

Study limitations should take into account that calculations of CO_2_E are approximate, and there is likely to be variation in the exact amount of GHG diverted as a result of this program. For example, while emissions resultant from transporting waste between the facility and the landfill are incorporated, potential GHG emissions from transporting surplus food to warehouses, food pantries, or households has not been factored in, nor does the model take into account any GHG emissions that may result from convention center attendees traveling to Las Vegas. Also, the volume of GHG that is generated from a section of landfill will dissipate over time or cease if the site is permanently capped off and waste is no longer added; alternatively, emissions may escalate as the landfill continues to grow [8]. Additionally, the number of meals created from the rescued food is an estimate. All meals served during the pilot project were congregate style, and servings were not measured for exact weights. Although the demand for surplus meals at the food pantries is very high, catering for 500–700 patrons per day, it cannot be guaranteed that each patron entirely consumed their meal [41]. Another consideration is that source reduction is a preferred method of food waste reduction over redistribution. After a scrutinizing examination of the amount of recoverable foods from convention centers, it is predicted that the amount of excess food will be decreased. While this may negatively impact the efforts to feed hungry and food insecure individuals, it is likely that the positive impact on climate change will be enhanced. 

This program is a downstream solution to food insecurity. In order for real progress to be made against food insecurity, individuals need education, employment with livable wages, and affordable health care. An investigation by Martin, Colantonio, Picho & Boyle (2016) explains that food-insecure families experience these challenges due to lack of confidence and inability to be self-sufficient [53]. Large and small organizations alike could do more to tackle food insecurity by helping secure at-risk populations in long-term employment, offering financial literacy programs to the public, or being involved in policy change that will encourage a decrease in food insecurity. Employers should also ensure that their employees are paid sufficient wages and provided access to healthcare.

Studies investigating the impact of the provision of prepared meals from recovered convention foods are limited. Aria’s convention food rescue program is unique in that it is the first of its kind at such a large scale and has the intentions to grow even larger. Alternatives like this program that better utilize recoverable food and divert food waste from landfills are important to human and environmental health.

## 5. Conclusions

Study findings indicated that roughly 45,383 meals were provided using food collected by this convention food rescue program between August 2016 and July 2017. According to a Hunger in America Client Survey, about 428,900 visits to the Three Square Food Bank’s meal program occur annually [51]. This means that meals produced from the convention food rescue program during this time period was enough to supply 10.6% of the meals provided by the congregate meal kitchens in Southern Nevada. This program is still in its beginning stages, and food rescue is only taking place at one convention center. As the program develops and becomes more efficient and more convention centers join the rescue efforts, the number of meals assembled and mouths fed could increase substantially given the number of conventions held annually in Las Vegas. The successful model for this rescue program should be mimicked in other similarly positioned cities around the world. 

There is an obligation to prevent the disposal of usable, edible food when many people are suffering from hunger and food insecurity. An increase in access to low- or no-cost meals for populations in need can decrease the harmful effects that being food-insecure can inflict. More meal availability can be predicted to decrease behavioral problems and increase school performance in children, decrease severity of chronic diseases in the elderly and people with disabilities, and reduce rates of stress and depression in adults. Also, as mentioned before, not only are mental and physical health outcomes effected by limited food, but financial, housing, hygiene, and health care needs are more likely to be met when food limitations are alleviated, all the more intensifying the benefits of diminishing food insecurity. 

Diverting food from landfills prevents the decomposing food waste from releasing GHG into the environment. Our findings revealed that 108 MT of CO_2_E were avoided by diverting recoverable food from the convention food rescue program in the pilot year away from the landfill. Findings are significant, as this food rescue program only included food from one convention center. As the convention food rescue program expands to include multiple convention properties, the prevention of GHG emissions will be amplified, and so too will the potential positive consequences on public and environmental health.

## Figures and Tables

**Figure 1 ijerph-16-01718-f001:**
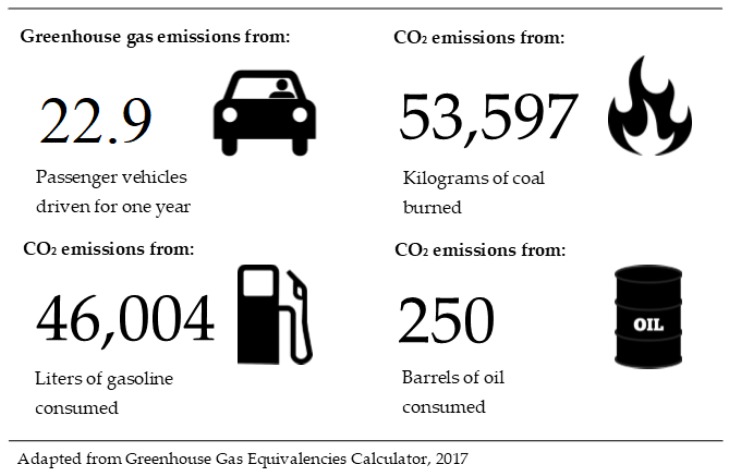
Examples of human related behaviors and actions that would have resulted in Greenhouse Gas Emissions of 108 MT CO_2_E [43].

**Table 1 ijerph-16-01718-t001:** WARM outputs and greenhouse gas emission proportions for the pilot year of Aria’s convention food rescue program [39].

GHG Emission	Proportions of CO_2_E (%)	Amount of GHG Emission (MTCO_2_E)
Carbon Dioxide	76.7	82.8
Methane	14.37	15.5
Nitrous Oxide	7.9	8.5
Total change in GHG Emissions (MTCO_2_E):		108

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
