# Peer review of "The Environmental Impact and Formation of Meals from the Pilot Year of a Las Vegas Convention Food Rescue Program"

_ijerph, 2019, doi:10.3390/ijerph16101718_

Round 1
Reviewer 1 Report
The paper is interesting because it deals with one aspect (food waste) which has numerous ethical and environmental implications.
I suggest the following changes:
i) The introduction should be reviewed, deepening the analysis of the environmental consequences of food waste and, above all, of the policies activated to reduce this phenomenon (also at the international level). Furthermore, the economic consequences of food waste should be also analyzed;
ii) The used methodology to calculate the amount of greenhouse gas (GHG) emissions should be explained in greater detail (par. 2.1. and 2.2.);
iii) The conclusions should include a more in-depth discussion of the strategies and policies that can be put in place to reduce food waste, avoiding repetition of issues that have already been dealt with.
Author Response
Response to Reviewer 1 Comments
Point 1: The introduction should be reviewed, deepening the analysis of the environmental consequences of food waste and, above all, of the policies activated to reduce this phenomenon (also at the international level). Furthermore, the economic consequences of food waste should be also analyzed;
Response 1: Thank you for this suggestion. A review of the environmental and economic consequences of food waste in the US and internationally have been added to the introduction. We have also added a discussion of policy implications and strategies to the discussion.
Point 2: The used methodology to calculate the amount of greenhouse gas (GHG) emissions should be explained in greater detail (par. 2.1. and 2.2.);
Response 2: Thank you for this suggestion. Increased detail on the Environmental Protection Agencies Waste Reduction Management program methodology has been added to the methods section. We have also added more detail on the information inserted into the warm model for estimated GHG outputs.
Point 3: The conclusions should include a more in-depth discussion of the strategies and policies that can be put in place to reduce food waste, avoiding repetition of issues that have already been dealt with
Response 3: Thank you for this suggestion. We have included a discussion of strategies and policies that can be used in conjunction with convention food recovery programs to reduce food waste and avoid repetition of issues.
Reviewer 2 Report
I have ignored minor typos for the copy-editing team
An interesting contribution and clearly depicts and analyses what is clearly an impressive effort; the technological capacities and scale of food banks to do this kind of food conveyance mirrors the sophistication of the commercial retail system in ways that bear testament to the extent of both waste and need in wealthy societies. I give some specific and more general comments below:
Lines 40-59: while some causes of food waste were briefly outlined, there is no mention made here of why and how people become food insecure ie income inequality; could you cite a reference or two?
60- is 'allowing' food to go to waste a practice? Arguably this is something that happens as a result of other practices, so is the moral charge of 'abhorrent' attributable to a human act, or the (often unseen/ignored) result of those acts. Wouldn't it be abhorrent even if people were not hungry, from an ecological perspective? And vice versa; we need to think how we would solve hunger if it were not for vast quantities of surplus food (there is a critical literature that sees it as abhorrent that poorer people are fed from the waste stream through charity, also)
64- surplus 'experienced by retailers'; this suggests that the actions and motives of retailers is not a cause of unnecessary surplus
76- I sense a gap in the explanation of the specific causes of convention food waste; authors give some reasons for how retailers cause wastage, but are there specific nodes in the generation of convention surplus that could, perhaps in a different article, be pinpointed as foci for waste-prevention efforts eg portion sizes, styles of service?
80-81: It would be interesting to consider if, and how much of the food gets wasted at the agency end, rather than assuming that all food is eaten.
99- Is it certain that the food would have been sent to landfill rather than other disposal options such as biogas digester, if it were not donated? Presumably not all leftover food is suitable for donation; what less harmful options than landfill could be used?
136- While not the focus of the article, there is a case to be made that a 'sustainable food system' would not involve thousands of people travelling (presumably by plane) to conference centres, a model which presumably makes it hard for organisers to plan how much food is required and potentially encourages significant over-preparation. Is there a risk that simply redirecting the wasted food enables the continuation of an excessive system? How could convention centre managers be engaged to prevent surpluses in first place? Or change their menus to involve lower-impact foods like less meat/dairy? I make this point simply to highlight the bigger-picture ways of thinking about the 'convention' food system, and that the GHG that go into the production/transportation of food that is then wasted presumably outstrips the savings made in donating it, pointing to the need to seek prevention options.
144- While the potential health benefits of enabling food access for those in need is framed speculatively, these cannot be demonstrated from the present study; what research would be needed to demonstrate this? For example, how easy is it for people in need to access that food, and what kinds of barriers might they face e.g. ineligibility, shame/stigma, lack of mobility to access meal programs. There is evidence that for some, accessing food charity does not result in 'positive changes in wellbeing' but can itself be stressful, especially as it is not always a guaranteed source of food. Was there any effort in the study to judge whether donated foods were indeed 'nutritious'? Could there be a decline in the nutritional or gastronomic quality of the food through the chilling/thawing/reheating process involved in redistribution?
153- This section comes across a little abruptly; perhaps better in the intro/literature section? Also, again there's a case to be made that the kind of tourism and travel represented by convention centres causes more GHG than the quantities mitigated through managing their waste, though beyond article's scope.
174-187: Again, gives impression that donation and landfill are only disposal options: is there no alternative regionally eg biogas generation?
185-187: Some authors view charitable feeding as preventing other ways of tackling food insecurity through ensuring adequate incomes eg Poppendieck's 'Sweet Charity'; donation programs should not be presented as the only, or primary, way to reduce food insecurity.
Author Response
Response to Reviewer 2 Comments
Point 1: Lines 40-59: while some causes of food waste were briefly outlined, there is no mention made here of why and how people become food insecure ie income inequality; could you cite a reference or two?
Response 1: Thank you for this suggestion. Elaboration on the common causes and risk factors of food insecurity in households has been added to the introduction.
Point 2: Line 60- is 'allowing' food to go to waste a practice? Arguably this is something that happens as a result of other practices, so is the moral charge of 'abhorrent' attributable to a human act, or the (often unseen/ignored) result of those acts. Wouldn't it be abhorrent even if people were not hungry, from an ecological perspective? And vice versa; we need to think how we would solve hunger if it were not for vast quantities of surplus food (there is a critical literature that sees it as abhorrent that poorer people are fed from the waste stream through charity, also)
Response 2: Thank you for bringing this to our attention. We have considered the way this has been phrased and adjusted it for clarification.
Point 3: Line 64- surplus 'experienced by retailers'; this suggests that the actions and motives of retailers is not a cause of unnecessary surplus
Response 3: Thank you for this suggestion. We have added additional detail to describe how actions and motives of retailers promote food waste.
Point 4: Line 76- I sense a gap in the explanation of the specific causes of convention food waste; authors give some reasons for how retailers cause wastage, but are there specific nodes in the generation of convention surplus that could, perhaps in a different article, be pinpointed as foci for waste-prevention efforts eg portion sizes, styles of service?
Response 4: Thank you for this suggestion. We have included details from program stakeholders that elaborate on their reason for so much extra food production in this section.
Point 5: Lines 80-81: It would be interesting to consider if, and how much of the food gets wasted at the agency end, rather than assuming that all food is eaten.
Response 5: Thank you for this suggestion. In the limitation section, we have included feedback from our program stakeholders that can comment on the receiving end of this program. While we cannot ensure that every meal is entirely consumed by food pantry clients, the demand for surplus meals by the community partners is very high.
Point 6: Line 99- Is it certain that the food would have been sent to landfill rather than other disposal options such as biogas digester, if it were not donated? Presumably not all leftover food is suitable for donation; what less harmful options than landfill could be used?
Response 6: Thank you for this question. We have included more information on the local landfill and related disposal methods. Yes, only food products that have been deemed suitable for this program gets rescued. Prior to this program, and while it has been in the beginning stages, the convention center continues to utilize these establishments.
Point 7: Line 136- While not the focus of the article, there is a case to be made that a 'sustainable food system' would not involve thousands of people travelling (presumably by plane) to conference centres, a model which presumably makes it hard for organisers to plan how much food is required and potentially encourages significant over-preparation. Is there a risk that simply redirecting the wasted food enables the continuation of an excessive system? How could convention centre managers be engaged to prevent surpluses in first place? Or change their menus to involve lower-impact foods like less meat/dairy? I make this point simply to highlight the bigger-picture ways of thinking about the 'convention' food system, and that the GHG that go into the production/transportation of food that is then wasted presumably outstrips the savings made in donating it, pointing to the need to seek prevention options.
Response 7: Thank you for drawing attention to this. The authors agree that while GHG cause by other sectors of the convention center is not the scope of the paper, it is worth a brief discussion. We have added this to the discussion section.
Point 8: Line 144- While the potential health benefits of enabling food access for those in need is framed speculatively, these cannot be demonstrated from the present study; what research would be needed to demonstrate this? For example, how easy is it for people in need to access that food, and what kinds of barriers might they face e.g. ineligibility, shame/stigma, lack of mobility to access meal programs. There is evidence that for some, accessing food charity does not result in 'positive changes in wellbeing' but can itself be stressful, especially as it is not always a guaranteed source of food. Was there any effort in the study to judge whether donated foods were indeed 'nutritious'? Could there be a decline in the nutritional or gastronomic quality of the food through the chilling/thawing/reheating process involved in redistribution?
Response 8: Thank you for this suggestion. We have included research from several publications that touches on the economic effects of food insecurity, as well as the changes in food-insecure households that utilize food pantries. Research and stakeholder input regarding nutritional quality of frozen/thawed meals was added the discussion section.
Point 9: Line 153- This section comes across a little abruptly; perhaps better in the intro/literature section? Also, again there's a case to be made that the kind of tourism and travel represented by convention centres causes more GHG than the quantities mitigated through managing their waste, though beyond article's scope.
Response 9: Thank you for this suggestion. The authors agree that moving this information into the introduction may be more appropriate. We have addressed quantities of GHG caused by other sectors of tourism in a response above.
Point 10: 174-187: Again, gives impression that donation and landfill are only disposal options: is there no alternative regionally eg biogas generation?
Response 10: Thank you again for this suggestion. This has been addressed in an above response, and details have been added into the discussion section.
Point 11: 185-187: Some authors view charitable feeding as preventing other ways of tackling food insecurity through ensuring adequate incomes eg Poppendieck's 'Sweet Charity'; donation programs should not be presented as the only, or primary, way to reduce food insecurity
Response 11: Thank you for bringing this to our attention. We have included suggestions that the convention center could assist with to decrease the prevalence of food insecurity in the area.
Round 2
Reviewer 1 Report
The description of the methodology has been improved, but it is still quite superficial, especially about to with concerning the Waste Reduction Model.
Some references should be checked (for example line 228, line 285).
Author Response
Response to Reviewer 1 Comments – Round 2
Point 1: The description of the methodology has been improved, but it is still quite superficial, especially about to with concerning the Waste Reduction Model.
Response 1: Thank you for this concern. We have added more detail, and when compared to other publications using the WRAM program, feel we have reached a similar level of explanation of the methodology that we can provide of the WARM program.
Point 2: Some references should be checked (for example line 228, line 285).
Response 2: Thank you for pointing this out, we have corrected

Reviewer 2 Report
Line 69-71: Thanks for including more references around explanatory factors for food insecurity. Reference [7] does not give the impression that it is lack of money/resources that accounts for food insecurity "sometimes" so this word should be removed; the majority of literature by experts on food insecurity, as well as USDA Food Security module itself, suggest that it is nearly always a lack of resources that causes food insecurity.
Lines 73-77: Ref [25] makes no references to homelessness, smoking, 'household head', visiting fathers or seasonality so I am unsure where these factors have been derived from. In any case, many of these potential risk factors are entwined with low income/poverty in the first place ('not receiving child support', racial groups' historical vulnerability to poverty, homelessness, high unemployment etc) so the wording is problematic: low income/inequality is core characteristic most of these determinants. Also, where is the evidence that 'lack of financial management skills' plays a significant role in food insecurity levels? Citations required
Line 79: "risking the health of themselves/children": wording sounds moralistic and as though people are making an active choice to reduce food spending, rather than this being an inevitable result of having an inadequate income, and food being a more flexible component of spending than bills/rent etc. Same with "placed less importance"; again, these 'choices' are usually prioritisations that people in low incomes are forced to make, rather than being a reflection of moral character as the wording seems to imply. Needs re-wording (the language in the original Feeding America surveys is more appropriate, even though these are based on food pantry users so will not be representative of all people experiencing food insecurity- other studies highlight the stigma of accessing food charity, reducing its effectiveness in addressing needs of all).
Line 96-98: Please provide evidence of this
Line 102: 'Surplus is EXPERIENCED by': similar to my earlier comment, this gives impression that surplus is an accidental outcome rather than something actively GENERATED by retain practices...seasonality etc is of course a factor but it is the overstocking an, often, poor matching of supply/demand that results in such huge amounts of wastage.
152-165: I presume the other reviewer asked for more specificity on methodology. It is beyond my specialism to assess this. However, though you mention methane recovery the landfill modelling assumes there are no alternative options for food waste disposal: is there no anaerobic digestion/composting capacity for Las Vegas? If so, these should be included in a comparison of GHG-emissions of different disposal options, including donation. These papers worth referencing: https://www.mdpi.com/2071-1050/7/4/4707/ and Phillips C, Hoenigman R, Higbee B, Reed T (2013) Understanding the Sustainability of Retail Food Recovery. PLoS ONE 8(10): e75530. doi:10.1371/ journal.pone.0075530. On the benefits of feeding surplus to animals, see work of zu Ergassen- https://www.researchgate.net/publication/324732630_Support_amongst_UK_pig_farmers_and_agricultural_stakeholders_for_the_use_of_food_losses_in_animal_feed and this from the UK: https://feedbackglobal.org/wp-content/uploads/2018/07/Pig-Idea-UK-policy-report.pdf
212-220: Still no evidence provided for these asserted benefits: while food insecurity clearly has any health implications, I would expect to see some evidence that redistributed surplus via charity is indeed an effective way of mitigating these problems. Instead of e.g. 'can decrease' (line 214), if no evidence can be provided, better to frame this as 'can be predicted/claimed to decrease'
222-223: I could see no mention of 'food supply programs' in ref [28]; the article rather mentions state entitlements such as SNAP, which is a very different way of bolstering food access than charitable food. This also applies to the following sentence: the implication that 'food supply programs' includes redistributed waste via charity is not demonstrated in the reference given.
231: yes, Gundersen and others argue for the impact of 'food assistance programs' but they refer to state entitlements such as SNAP, not, as your article is about, charitable food redistribution, so the use of these articles to support the health benefits claimed by the meal rescue program is not appropriate. GHG savings may be evident, but I'm not convinced about the impact of the program on the health and dignity of those receiving the meals.
235-240: these points are valid but again, where is the evidence that these benefits can be brought about through meal rescue?
257-265: I would be very interested in evidence that the meal rescue program prevented the amount of surplus meals by the convention centre: what kind of prevention approaches could they employ e.g around portion sizes/meal formats?
283-290: YES!
Author Response
Response to Reviewer 2 Comments – Round 2
Point 1: Line 69-71: Thanks for including more references around explanatory factors for food insecurity. Reference [7] does not give the impression that it is lack of money/resources that accounts for food insecurity "sometimes" so this word should be removed; the majority of literature by experts on food insecurity, as well as USDA Food Security module itself, suggest that it is nearly always a lack of resources that causes food insecurity.
Response 1: Thank you for this suggestion. We have changed the language to express a stronger position.
Point 2: Lines 73-77: Ref [25] makes no references to homelessness, smoking, 'household head', visiting fathers or seasonality so I am unsure where these factors have been derived from. In any case, many of these potential risk factors are entwined with low income/poverty in the first place ('not receiving child support', racial groups' historical vulnerability to poverty, homelessness, high unemployment etc) so the wording is problematic: low income/inequality is core characteristic most of these determinants. Also, where is the evidence that 'lack of financial management skills' plays a significant role in food insecurity levels? Citations required
Response 2: Thank you for this comment. We have verified that Gundersen, et al. have discussed the mentioned risks. The factors associated with food insecurity can be found beginning on page 376 of the publication.
Point 3: Line 79: "risking the health of themselves/children": wording sounds moralistic and as though people are making an active choice to reduce food spending, rather than this being an inevitable result of having an inadequate income, and food being a more flexible component of spending than bills/rent etc. Same with "placed less importance"; again, these 'choices' are usually prioritisations that people in low incomes are forced to make, rather than being a reflection of moral character as the wording seems to imply. Needs re-wording (the language in the original Feeding America surveys is more appropriate, even though these are based on food pantry users so will not be representative of all people experiencing food insecurity- other studies highlight the stigma of accessing food charity, reducing its effectiveness in addressing needs of all).
Response 3: Thank you for this suggestion. We have reworded this section and corrected reference
Point 4: Line 96-98: Please provide evidence of this
Response 4: Thank you for this suggestion. The purpose of this sentence was to lead into the next topic. We have adjusted the phrasing to reflect this.
Point 5: Line 102: 'Surplus is EXPERIENCED by': similar to my earlier comment, this gives impression that surplus is an accidental outcome rather than something actively GENERATED by retain practices...seasonality etc is of course a factor but it is the overstocking an, often, poor matching of supply/demand that results in such huge amounts of wastage
Response 5: Thank you for this suggestion. We have changed to the phrasing to indicate the active waste practices.
Point 6: 152-165: I presume the other reviewer asked for more specificity on methodology. It is beyond my specialism to assess this. However, though you mention methane recovery the landfill modelling assumes there are no alternative options for food waste disposal: is there no anaerobic digestion/composting capacity for Las Vegas? If so, these should be included in a comparison of GHG-emissions of different disposal options, including donation. These papers worth referencing: https://www.mdpi.com/2071-1050/7/4/4707/ and Phillips C, Hoenigman R, Higbee B, Reed T (2013) Understanding the Sustainability of Retail Food Recovery. PLoS ONE 8(10): e75530. doi:10.1371/ journal.pone.0075530. On the benefits of feeding surplus to animals, see work of zu Ergassen- https://www.researchgate.net/publication/324732630_Support_amongst_UK_pig_farmers_and_agricultural_stakeholders_for_the_use_of_food_losses_in_animal_feed and this from the UK: https://feedbackglobal.org/wp-content/uploads/2018/07/Pig-Idea-UK-policy-report.pdf
Response 6: Thank you for this suggestion. The nearby landfill is not currently offering composting options, primarily due to the lack of investment interest in the practice. There are up-and-coming private groups that are composting in the area, but have not connected yet with large, local businesses. We have added additional detail of your suggested topics in the discussion.
Point 7: 212-220: Still no evidence provided for these asserted benefits: while food insecurity clearly has any health implications, I would expect to see some evidence that redistributed surplus via charity is indeed an effective way of mitigating these problems. Instead of e.g. 'can decrease' (line 214), if no evidence can be provided, better to frame this as 'can be predicted/claimed to decrease'
Response 7: Thank you for this suggestion. As the evidence regarding food insecurity and poor health outcomes goes primarily in one direction, our phrasing has been adjusted to indicate the potential that being food secure can have on health and well-being.
Point 8: 222-223: I could see no mention of 'food supply programs' in ref [28]; the article rather mentions state entitlements such as SNAP, which is a very different way of bolstering food access than charitable food. This also applies to the following sentence: the implication that 'food supply programs' includes redistributed waste via charity is not demonstrated in the reference given.
Response 8: Thank you for this suggestion. We have rearranged statements, added additional details, and rephrased some discussions in this section.
Point 9: 231: yes, Gundersen and others argue for the impact of 'food assistance programs' but they refer to state entitlements such as SNAP, not, as your article is about, charitable food redistribution, so the use of these articles to support the health benefits claimed by the meal rescue program is not appropriate. GHG savings may be evident, but I'm not convinced about the impact of the program on the health and dignity of those receiving the meals.
Response 9: Thank you for this suggestion. We have rearranged statements, added additional details, and rephrased some discussions in this section.
Point 10: 235-240: these points are valid but again, where is the evidence that these benefits can be brought about through meal rescue?
Response 10: Thank you for this suggestion. We have adjusted our statements to discuss how meal rescue and increased access to meals change a household’s economic redistribution changing social and health outcomes.
Point 11: 257-265: I would be very interested in evidence that the meal rescue program prevented the amount of surplus meals by the convention centre: what kind of prevention approaches could they employ e.g around portion sizes/meal formats?
Response 11: Thank you for this comment. We have elaborated on approaches that the convention center can do to prevent waste.
Point 12: 283-290: YES!
Response 12: Thank you very much for your time.
